ecology, evolution, physiology

phenotypic plasticity, reaction norms, bupropion

**Author for correspondence:**
Semona Issa
e-mail: semona.issa@ntnu.no

# Dopamine mediates life-history responses to food abundance in *Daphnia*

Semona Issa[1], Marlène Gamelon[1], Tomasz Maciej Ciesielski[2], Kristine Vike-Jonas[3], Alexandros G. Asimakopoulos[3], Veerle L. B. Jaspers[2] and Sigurd Einum[1]

[1]Centre for Biodiversity Dynamics (CBD), Department of Biology, [2]Department of Biology, and [3]Department of Chemistry, Norwegian University of Science and Technology, Høgskoleringen 5, 7491 Trondheim, Norway

SI, 0000-0002-5628-2516; MG, 0000-0002-9433-2369; SE, 0000-0002-3788-7800

Expression of adaptive reaction norms of life-history traits to spatio-temporal variation in food availability is crucial for individual fitness. Yet little is known about the neural signalling mechanisms underlying these reaction norms. Previous studies suggest a role for the dopamine system in regulating behavioural and morphological responses to food across a wide range of taxa. We tested whether this neural signalling system also regulates life-history reaction norms by exposing the zooplankton *Daphnia magna* to both dopamine and the dopamine reuptake inhibitor bupropion, an antidepressant that enters aquatic environments via various pathways. We recorded a range of life-history traits across two food levels. Both treatments induced changes to the life-history reaction norm slopes. These were due to the effects of the treatments being more pronounced at restricted food ration, where controls had lower somatic growth rates, higher age and larger size at maturation. This translated into a higher population growth rate ($r$) of dopamine and bupropion treatments when food was restricted. Our findings show that the dopamine system is an important regulatory mechanism underlying life-history trait responses to food abundance and that bupropion can strongly influence the life history of aquatic species such as *D. magna*. We discuss why *D. magna* do not evolve towards higher endogenous dopamine levels despite the apparent fitness benefits.

# 1. Introduction

Phenotypic plasticity is the propensity of a genotype to produce different phenotypes across environments [1,2]. Under natural selection, the slope and elevation of the relationship between environment and phenotype (i.e. the reaction norm) can evolve such that it approaches optimality with respect to fitness [3]. In this case, the reaction norm is adaptive since it gives higher fitness in each environment than any alternative reaction norm [4–6]. Expression of adaptive reaction norms is therefore crucial to maintain high fitness in environments that vary across space and time. One of the environmental factors that shows extensive spatial and temporal variation is food availability. Expression of reaction norms to food availability includes allocation patterns to different components of the life history such as reproduction, growth and somatic maintenance [7–9]. For example, resource allocation to somatic maintenance (survival) increases at the cost of growth and reproduction under food limitation in short-lived species [7,10–12]. Thus, expressing appropriate reaction norms for different life-history traits in response to food availability has important fitness consequences.

At the molecular level, the responses to environmental stimuli that produce these reaction norms are mediated by neural signalling mechanisms. Thus, knowledge about these mechanisms is important to understand how organisms adjust their phenotypes under environmental change (see [13] for a review on

neuronal pathways involved in phenotypic plasticity). For the specific case of food abundance, the neurotransmitter dopamine, which is synthesized by most animals, has been shown to play an important role in modulating behavioural and morphological responses. In the nematode *C. elegans*, dopamine is released from dopaminergic neurons when food is present [14], causing a reduction in the animal's rate of locomotion [15] and possibly regulating their lifespan [14]. In mammals, obese individuals release more dopamine upon food consumption and hence experience a higher reward sensation from food intake compared with lean individuals [16]. In honeybees (*Apis mellifera carnica*) and *Drosophila* larvae, dopamine is involved in learning to associate food odour with aversiveness of taste, and thereby mediates an avoidance behaviour towards toxic and/or unpalatable food [17,18]. In sea urchin larvae (*Strongylocentrotus purpuratus*), dopamine reduces food acquisition through a shortening in arm length when food is abundant, which preserves energy that can be allocated to other functions [19]. Hence, the dopamine system appears to be tightly linked to the regulation of food responses and may therefore be a likely candidate neural signalling system regulating the life-history reaction norms. If so, chemically induced changes to dopamine levels are predicted to change the slopes of these reaction norms.

Insights into the mediation of reaction norms by neurotransmitters are also of potential value for environmental risk assessment of pharmaceutical products. Specifically, in aquatic biota, neurotransmitter systems can be directly altered by anthropogenic activity through environmental release of antidepressants. Following administration to humans, antidepressants can be eliminated unmetabolized or as active metabolites and enter the aquatic environment through wastewater [20]. Another path by which pharmaceuticals can enter the aquatic environment is by the disposal of unused products [21]. Exposure to released pharmaceuticals can influence the behaviour, development, reproduction and survival of fish, invertebrates and amphibians [22,23]. Hence, more research on ecological effects of antidepressants in aquatic habitats is needed, as these can impact individual fitness and population viability [23]. Furthermore, interactive effects between antidepressant disruption of neurotransmitter systems and environmental variables such as food abundance can be expected. Of particular interest in the case of the dopamine system is bupropion, which is used both as an antidepressant and as treatment for smoking cessation [24]. Bupropion inhibits the neuronal reuptake of norepinephrine and dopamine, increasing their concentration in the synaptic cleft [25]. Bupropion has previously been detected in natural surface water, stream sediments as well as in fish brain tissue [26,27], and has been shown to affect the physiology, morphology and behaviour of aquatic animals. For example, bupropion can alter the morphology and predator avoidance behaviour of fathead minnows (*Pimephales promelas*), as well as directly affect their survival [28,29]. Hence, if dopamine is indeed involved in regulating life-history reaction norms in response to food abundance, then disruption of the dopamine system by bupropion is expected to lead to changes in the slopes of these reaction norms.

In this study, we experimentally tested for the effects of dopamine and bupropion exposure on the reaction norms of life-history traits of *Daphnia magna* in response to high versus restricted food ration. *Daphnia* are keystone zooplankton in freshwater ecosystems and model organisms for studying

anthropogenic and natural stressors in these ecosystems [30]. They have also been used in studies of the dopamine signalling system [31,32]. We hypothesize that *D. magna* with natural dopamine levels will have life-history reaction norms that approach optimality with respect to fitness in response to food abundance, and that disruption of these levels will lead to a change in the response to food abundance and hence the slopes of these reaction norms. Furthermore, bupropion administration causes an increase in extracellular dopamine in the brain [33]. Thus, if this is the dominating effect of this treatment, we expect dopamine and bupropion exposure to induce similar changes to the slopes of these reaction norms relative to the control treatment.

## 2. Material and methods

### (a) Study organisms

Ephippia containing resting eggs resulting from sexual reproduction of *D. magna* were collected in November 2014, in a pond at Værøy Island (1.0 ha, 67.687°N 12.672°E), northern Norway. Ephippial eggs were hatched in the laboratory and propagated clonally. For this experiment, juveniles of a single clone (clone 47) of *D. magna* were asexually propagated for four successive generations prior to use. A maximum of 30 individuals of *D. magna* were cultured in 2.5 l aquaria at 20°C in a modified Aachener Daphnien Medium (ADaM) [34] (SeO$_2$ concentration reduced by 50%), under long photoperiods (16 h L : 8 h D) using white fluorescent lamps. The medium was exchanged weekly and the animals were fed three times a week with Shellfish Diet 1800 (Reed Mariculture Inc.) at a final concentration of $3.2 \times 10^5$ cells ml$^{-1}$.

### (b) Experimental design

A full factorial design with control, dopamine, bupropion and two food rations (high versus restricted) was used, with thirty 50 ml replicate tubes for each of the six combinations (electronic supplementary material, figure A1). Aqueous exposure to dopamine allows us to directly manipulate this compound in the experimental organisms. The exposure concentration of dopamine (2.3 mg l$^{-1}$) was chosen for successfully inducing changes in *D. magna* growth based on a study by Weiss *et al*. [35], and that of bupropion (1 µg l$^{-1}$) was selected for being an environmentally relevant concentration that can be expected to influence life-history traits based on a pilot study we conducted prior to this experiment (see [36] and electronic supplementary material, figures A2 and A3). Bupropion stock solutions (0.0016 g l$^{-1}$) were prepared by dissolving bupropion hydrochloride (Sigma-Aldrich, St Louis, MO, USA) in ultrapure water (18.2 MΩ cm; Milli-Q Plus, Millipore Corp.). The stock solutions were then added to ADaM to create the desired bupropion exposure concentration. For the dopamine treatment, dopamine hydrochloride (Sigma-Aldrich, St Louis, MO, USA) was first dissolved in 100 ml ultrapure water before dilution in ADaM to the desired exposure concentration. Controls containing only ADaM medium were performed parallel to the exposure replicates.

For each replicate tube, a single female neonate (less than 24 h old) was introduced and kept at 20°C under long photoperiods (16 h L: 8 h D) until death. The medium was renewed in all replicates ($n = 180$) three times a week, and the animals were fed at each renewal event with Shellfish Diet 1800 at a final concentration of $2.88 \times 10^5$ cells ml$^{-1}$ (*ad lib* at 20°C) for the high food ration and $8.6 \times 10^4$ cells ml$^{-1}$ (30% *ad lib* at 20°C) for the restricted food ration. Day 0 marks the start of the experiment, which was completed when the last individual died. Male individuals ($n = 9$) and individuals that died from pipetting ($n = 2$) were removed and not replaced.

## (c) Sampling procedure and measurements of life-history traits

Conductivity (WTW LF 330 conductivity metre), pH (WTW pH 340i) and dissolved oxygen (WTW Multi 3410 multiprobe metre) were measured throughout the experiment, after medium renewal, in the new exposure solutions and ADaM medium used for the controls ($n = 27$; nine samples collected in total from each of the dopamine, bupropion and control treatments). Simultaneously, the new exposure solutions and ADaM medium were sampled for bupropion and dopamine analysis ($n = 21$; seven samples collected in total from each of the dopamine, bupropion and control treatments).

The samples were stored at −20°C for a maximum of four months after collection, prior to analysis. Two complementary sample preparation protocols were employed to cover all concentration ranges: (i) dilute-and-shoot and (ii) liquid–liquid extraction. Subsequent analysis was performed by ultra-performance liquid chromatography coupled to a triple quadrupole mass analyser (UPLC-MS/MS). Further details on the method are provided in electronic supplementary material. Over the course of the experiment, pH, conductivity and dissolved oxygen were within the recommended range for testing of chemicals in *D. magna*, according to OECD guidelines [37]. The conductivity remained at 1.1 mS/cm, mean dissolved oxygen at 9.0 mg l$^{-1}$ and pH at 8.3 across treatments, whereas measured average concentrations of dopamine and bupropion were within 13% and 10% of their nominal concentrations, respectively (electronic supplementary material, table A2). Lower than expected concentrations of these compounds may have been caused by degradation during storage.

Immediately prior to exposure on day 0, neonates were photographed for body length measurements (BL, mm, measured from the upper margin of the eye to the junction of the carapace and spine) using ImageJ v. 1.52a (National Institutes of Health, Bethesda, MD). BL measurements were then transformed to dry mass (DM, mg) using the equation by Yashchenko *et al.* [38]: $DM = 0.00535 \times BL^{2.72}$. Thereafter, individual replicates were checked daily to record age at maturation and age at second reproduction, defined as the time when eggs were first visible in the brood chamber. Body length at first reproduction was also measured using ImageJ as described above. Live progeny released were collected and counted to yield first and second clutch size. For each replicate, we measured the BL of three offspring that were randomly sampled from each of the first and second clutch. As a derived parameter, we calculated the first clutch biomass as the product of clutch size and average offspring DM for that clutch. Offspring from all subsequent clutches were removed at each medium change, and the longevity of the mothers was recorded.

The somatic growth rate (SGR) of each replicate was calculated using the equation

$$SGR = \frac{\ln(DM_{end}) - \ln(DM_{start})}{duration}, \tag{2.1}$$

where $DM_{start}$ is the dry mass (in mg) of the replicate at the neonatal stage, $DM_{end}$ is the dry mass (in mg) of the replicate at maturation and duration is the number of days between the two stages.

The intrinsic population growth rate ($r$) was calculated based on the two first reproductive events from the Euler–Lotka equation,

$$\sum_{x=0}^{\infty} l_x m_x e^{-rx} - 1 = 0, \tag{2.2}$$

where age $x$ can be either age at maturation or age at second reproduction, $l_x$ is the probability of survival to age $x$ and $m_x$ is the average number of offspring produced by an individual of age $x$.

## (d) Statistical analyses

All statistical analyses and graphic illustrations were performed in R v. 3.5.2. [39]. We first tested whether the slopes of the reaction norms of the measured life-history traits in response to food abundance differed among treatments. To assess this for DM at maturation and SGR, we used generalized least-squares regression (GLS) models including the effects of the two categorical predictors, treatment and food (high versus restricted) and their interaction. For offspring DM (first and second clutch analysed separately), linear mixed effects (LME) models were fitted with treatment, food and their interaction as fixed predictor variables and replicate as a random predictor variable. We also tested the effects of treatment, food and their interaction on clutch size, age at maturation, age at second reproduction and longevity, using Poisson generalized linear models (Poisson GLMs).

Model selection followed a backwards selection procedure, where variables were removed sequentially, starting with random effects, using likelihood ratio tests [40]. For GLS and LME models, residuals were checked for homogeneous variance and for normal distribution. The VarIdent command from the *nlme* package was used to allow residual variance to differ among treatments and food (see [41] for an example using a variance function [42]). Poisson GLM models were tested for overdispersion and their Pearson and deviance residuals were checked for patterns and lack of fit. To deal with overdispersion for models for age at maturation and longevity, we used a quasi-Poisson GLM instead of a Poisson GLM. Tukey's multiple comparison test was implemented where groups were significantly different. For intrinsic population growth rates ($r$), bootstrapped sample means were used to compute $r$ values for which 95% confidence intervals were derived using the percentile method. Between-group differences in $r$ were considered statistically significant in the case where 95% confidence intervals did not overlap. The models were implemented using the *lme* and *gls* functions in the package *nlme* [43] and the *glm* function in the package *stats*.

To determine the causal pathways from food ration to first clutch biomass through age and DM at maturation, we used confirmatory path analyses [44,45]. Because we expected the causal path model to be the same for the three treatments (dopamine, bupropion and control) but the relationships between life-history traits to differ in terms of strength and/or direction among treatments, we fitted a model of hypothesized paths between traits, which we applied separately for each of the control, bupropion and dopamine datasets. This path model consisted of a sequence of linear regressions where food ration was used as a main effect explaining the variation in the different traits. Note, however, that an interaction between food ration and age at maturation was added in the path model for the bupropion treatment (see results). For each linear regression, we recovered both standardized and unstandardized regression coefficients and their SE. The overall goodness-of-fit of the models was assessed using Shipley's test of directional separation which yields a chi-squared distributed Fisher's C statistic. A $p > 0.05$ indicates that no significant paths are missing from the model and that it fits the dataset well [44]. The paths models were implemented using the *piecewiseSEM* package [46].

## 3. Results

## (a) Reaction norms in response to food ration

For all traits except longevity, the reaction norm slopes in response to food ration were of the same sign for the dopamine, bupropion and control treatments. This indicates that life-history traits responded in the same direction to a change in food ration, irrespective of the treatment. For all treatments, SGR, first and second clutch size increased with higher food ration ($p < 0.001$), whereas age at maturation, DM at

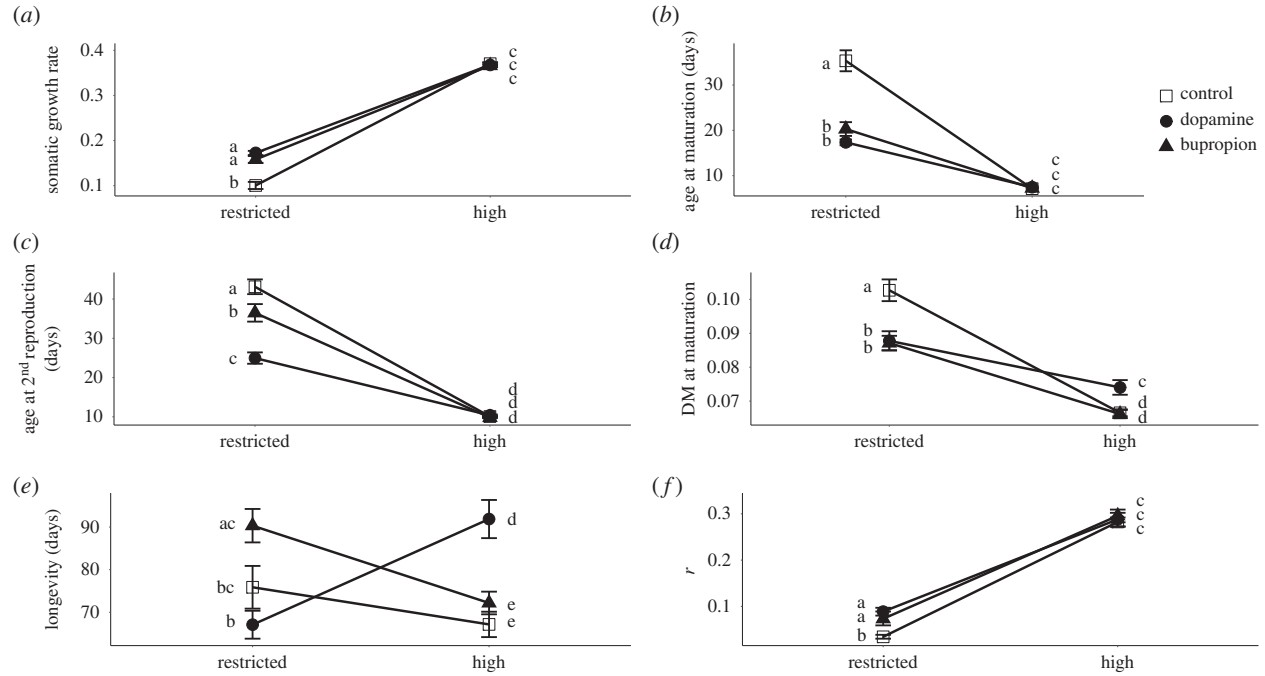

**Figure 1.** Effect of food ration on maternal traits in *D. magna* in the dopamine, bupropion and control treatments. (*a*) Somatic growth rate, (*b*) age at maturation (days), (*c*) age at second reproduction (days), (*d*) dry mass at maturation (mg), (*e*) longevity (days) and (*f*) intrinsic population growth rate (*r*). Error bars give 1 s.e. for (*a*–*e*) and 95% CI for (*f*). Means with the same letter are not significantly different from each other.

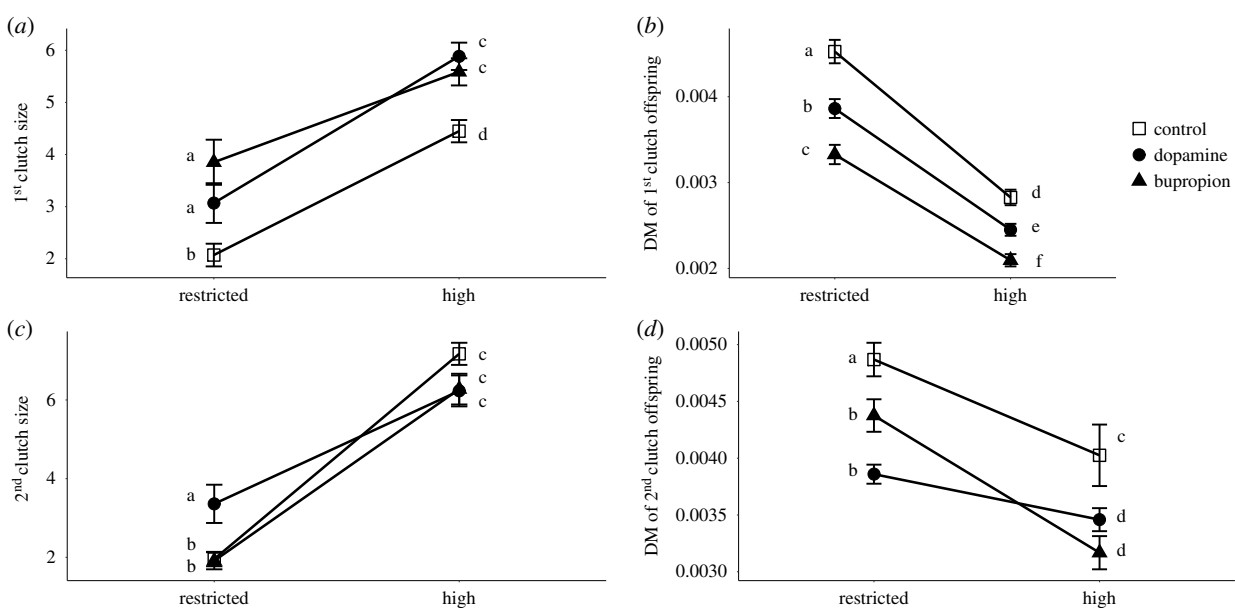

**Figure 2.** Effect of food ration on offspring production traits in *D. magna* in the dopamine, bupropion and control treatments. (*a*) First clutch size, (*b*) dry mass of first clutch offspring (mg), (*c*) second clutch size and (*d*) dry mass of second clutch offspring (mg). Error bars give 1 s.e. Means with the same letter are not significantly different from each other.

maturation, DM of first and second clutch and age at second reproduction decreased when the resources became sufficient ($p < 0.01$) (figures 1 and 2). For longevity, food restriction tended to increase it in both control and bupropion treatments (ns for control treatment, $p < 0.05$ for bupropion treatment), whereas the opposite pattern was observed in the dopamine treatment ($p < 0.001$). Although the sign of the reaction norm (i.e. positive versus negative slope) did not depend on the exposure treatment for most traits, their steepness did (for model selection results see electronic supplementary material, table A3). This was generally due to a more pronounced effect of dopamine and bupropion under restricted than under high food regimes (figures 1 and 2).

Specifically, at high food ration, treatment had no effect for SGR, age at maturation and age at second reproduction (ns), whereas a strong effect of dopamine treatment was observed for DM at first reproduction ($p < 0.01$). By contrast, at restricted food ration, the differences between control on one hand and dopamine and bupropion treatments on the other hand, became more pronounced ($p < 0.01$) (figure 1*a*–*d*; electronic supplementary material, table A5). Moreover, exposure to dopamine and bupropion induced lower DM for first and second clutch compared to controls, independent of food level ($p < 0.01$) (figure 2; electronic supplementary material, table A5). Finally, whereas the effects of food ration on life-history traits described above translated into an expected

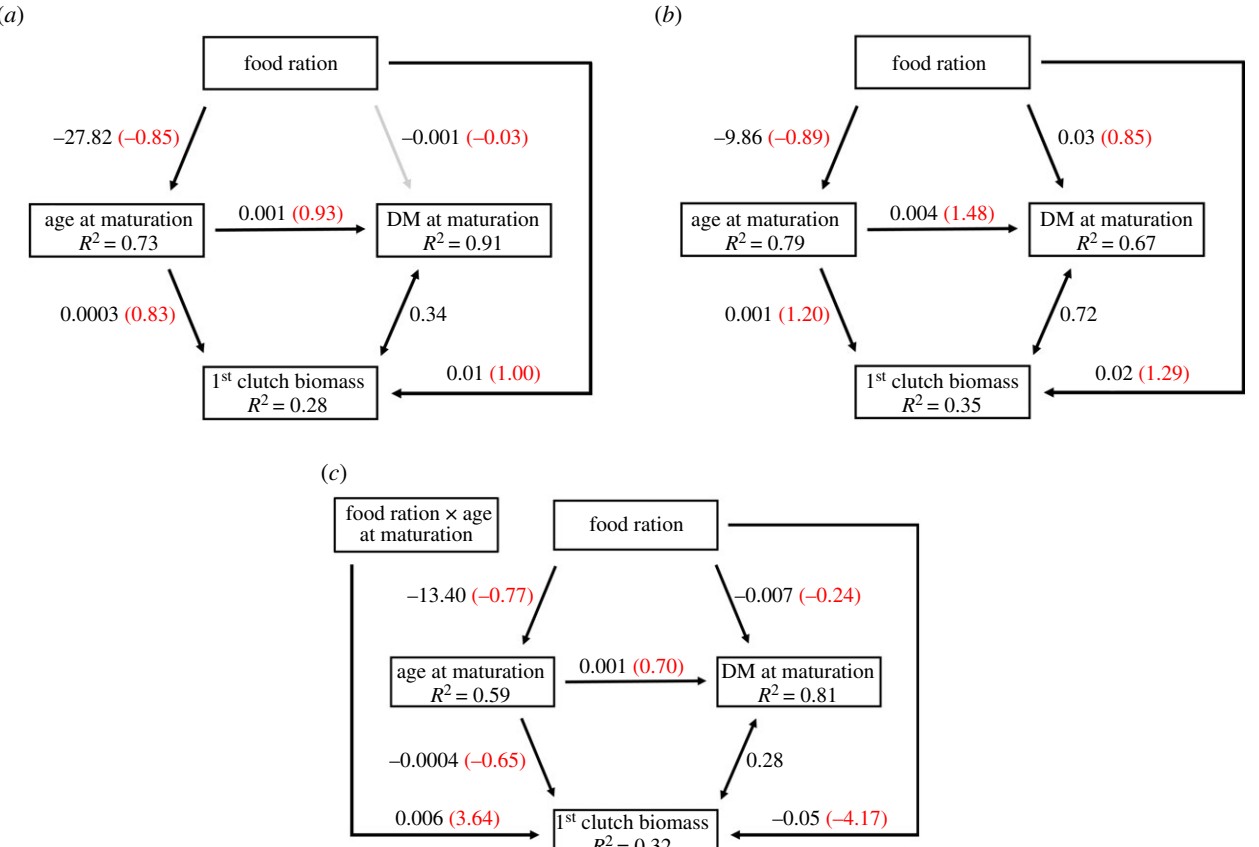

**Figure 3.** Structural equation models (SEM) exploring the effects of food ration on age at maturation, DM at maturation and first clutch biomass and the relationships between these across the (*a*) control, (*b*) dopamine and (*c*) bupropion treatments. Single-headed arrows represent unidirectional relationships among variables while double-headed arrows represent correlated errors between two dependent variables. Arrow for non-significant path ($p \geq 0.05$) is shown in grey. $R^2$ for component models are given in the boxes of response variables. Standardized coefficients, obtained by scaling the coefficients $\beta$ by the ratio of the standard deviation of *x* over the standard deviation of *y*, are given in red (in parentheses) and unstandardized coefficients are given in black. (Online version in colour.)

strong decline in population growth rate (*r*) at restricted food, this effect was steeper for the control than for the dopamine and bupropion treatments ($p < 0.05$). Under restricted food ration, *r* was 159% and 114% higher in the dopamine and bupropion treatments than in the control, respectively (figure 1*f*; electronic supplementary material, table A5).

## (b) Relationships among life-history traits at different food rations

The path models for the exposure treatments and the control fit the datasets very well ($p = 1$ for all groups). A high food ration favoured earlier maturation ($p < 0.05$). The direct effect of food on age was however much smaller in magnitude in the dopamine and bupropion treatments compared to the control ($\beta_{control} = -27.82 \pm 2.3$, $\beta_{dopamine} = -9.86 \pm 0.7$ and $\beta_{bupropion} = -13.40 \pm 1.6$, unstandardized coefficients; figure 3). In turn, age at maturation was positively associated with DM at maturation that was itself positively correlated with first clutch biomass ($p < 0.05$) (figure 3). DM at maturation was also affected by food ration directly (ns for control treatment, $p < 0.05$ for dopamine and bupropion treatments), but this effect was of smaller magnitude ($\beta_{control} = -0.03$, $\beta_{dopamine} = 0.85$ and $\beta_{bupropion} = -0.24$, standardized coefficients) than its indirect effect (through age at maturation), which is obtained by multiplying the path coefficients ($\beta_{control} = -0.85 \times 0.93 = -0.79$, $\beta_{dopamine} = -1.32$ and $\beta_{bupropion} = -0.54$, standardized coefficients; figure 3).

Direct effects of age at maturation and food ration on first clutch biomass were observed in addition to the positive correlation with DM at maturation ($p < 0.05$). In the control and dopamine treatments, the direct effect of age at maturation on biomass was positive, whereas it was negative in the bupropion treatment (figure 3). This negative effect was nonetheless weaker under high food ration ($\beta_{food\ ration \times age\ at\ maturation} > 0$, $p < 0.001$; figure 3*c*; electronic supplementary material, figure A4). Furthermore, the direct effect of food on biomass was larger in magnitude than its indirect effect ($\beta_{direct} = 1.00$ versus $\beta_{indirect} = -0.71$ in control; $\beta_{direct} = 1.29$ versus $\beta_{indirect} = -1.07$ in dopamine; $\beta_{direct} = -4.17$ versus $\beta_{indirect} = 1.82$ in bupropion, standardized coefficients; figure 3).

## 4. Discussion

In this study, we examined how dopamine mediates the responses of life-history traits to food abundance in *D. magna*, through aqueous exposure to dopamine and the antidepressant bupropion, a dopamine reuptake inhibitor. As hypothesized based on previous studies documenting behavioural and morphological effects of dopamine, dopamine and bupropion treatments significantly changed the slopes of life-history reaction norms to food abundance. The changes in slopes were due to effects of the treatments being more pronounced at a restricted food ration.

Life-history reaction norms to food abundance in the controls were consistent with previous empirical and theoretical

studies. Specifically, somatic growth rate decreased at restricted food ration thereby delaying maturation [47,48]. In turn, delayed maturation resulted in an increase in adult size (measured as DM at maturation). A larger size at maturity is believed to be metabolically advantageous, as it lowers the threshold food concentration at which assimilation equals respiration, making larger individuals able to grow and reproduce at lower food levels compared to smaller individuals [47]. This is caused by larger individuals having higher filtering rates than smaller individuals [49] and consequently higher feeding rates [50], which increases food uptake at low food concentrations. Once the threshold size is reached, energy can be allocated to reproduction [51,52]. Therefore, at restricted food ration, a higher somatic investment (adult size) at the expense of early life reproduction is likely to be adaptive, in line with resource allocation theory [12,53]. A similar argument can be made for an adaptive role of the observed reaction norm in terms of offspring size. At restricted food ration, offspring size increased whereas offspring number decreased. This trade-off between offspring size and number is due to energy limitations [54,55]. The optimal solution to this trade-off depends in turn on the food abundance [56]. Since the ability to support metabolic requirements at low food concentrations increases with body size in *Daphnia*, larger offspring have higher chances of surviving starvation [57]. Thus, mothers allocate their energy towards few but large offspring at low food conditions [58,59].

Although growth, somatic investment and reproduction responded qualitatively in the same way to food ration across treatments, quantitative differences were observed. This supports the view expressed above that these reaction norms to food abundance are under active physiological control and hence can respond to selection in an adaptive way, rather than being passive outcomes of energy availability. If observed differences between high and restricted food rations were solely based on the amount of energy available at each food ration, there would be no difference observed between the treatments at a given food ration.

At restricted food, under dopamine and bupropion exposure, resource allocation to maturation increased, leading to accelerated somatic growth rates, smaller adults, earlier ages at maturity and eventually shorter generation times (i.e. mean age of mothers) compared to the control. A positive effect of dopamine upregulation on somatic growth rate was also seen in Weiss *et al*. [35], who suspected it could be due to an effect of dopamine on cell proliferation and/or cell volume. In addition to accelerating growth, dopamine upregulation can stress organisms by exacerbating dopamine autoxidation, which produces reactive oxygen species and neurotoxins that damage dopaminergic neurons and cause oxidative stress [60,61]. Evidence for this may lie in the observed shorter generation times in the exposure treatments compared to the control. Indeed, several empirical studies have shown that fast species might exhibit accelerated life histories in response to stressful environmental conditions by reproducing earlier and accelerating their turnover [62,63]. Accordingly, we found that *D. magna*, a fast species, exhibits a faster pace of life under dopamine and bupropion exposure at the expense of adult size and offspring size. The smaller mothers in the exposure treatments produced smaller offspring, as can be expected from the known positive correlation between offspring size and mother size [64,65].

Despite the similar effects of bupropion and dopamine treatments on life-history reaction norms to food abundance,

path analyses identified differences in their resource allocation responses. Specifically, the relative importance of direct and indirect resource allocation (through age at maturation) to reproduction (first clutch biomass) changed according to food abundance across treatments. In the control and dopamine treatments, indirect resource allocation to reproduction increased at restricted food ration while direct allocation decreased ($\beta_{direct} > 0$ and $\beta_{indirect} < 0$ for both treatments). The opposite was true in the bupropion treatment ($\beta_{direct} < 0$ and $\beta_{indirect} > 0$). Moreover, direct allocation at restricted food ration was, given its magnitude, sufficient to offset the negative effect of delayed maturation on clutch biomass seen in the bupropion treatment. The negative effect of delayed maturation on clutch biomass in the bupropion treatment was unexpected, given the positive association between adult age, adult size and ultimately offspring size, and it could be due to physiological disruptions specific to bupropion's mode of action. Previous studies on aquatic animals have reported a variety of negative effects of bupropion exposure on reproductive physiology and development. One study showed bupropion negatively affecting the testicular morphology and reproductive physiology of adult male fathead minnows [29]. Another study reported disruption of zebrafish (*Danio rerio*) development, as well as a disruption of enzymatic activity related to energy production, movement and detoxification [66]. Finding differences in the resource allocation strategies of aqueous dopamine and bupropion was surprising, given that they were expected to have similar effects on the dopamine system and hence produce comparable physiological changes. However, aqueous dopamine and bupropion may be differently metabolized upon uptake and thus differ in their mechanisms of action and effects.

Regardless of their mode of action, aqueous dopamine and bupropion induced similar changes with respect to population growth rates (*r*). At restricted food ration, both treatments caused an increase in population growth rate (*r*). Individuals in these treatments allocated more resources to maturation and reproduction, advancing the timing of reproduction, which resulted in faster rates of population growth compared to the control. This boost in fitness did not induce any apparent long-term costs as longevity did not differ significantly across treatments at restricted food. This is an important finding as both the principle of allocation [9] and the disposable soma theories [67] predict reduced longevity as a consequence of a greater allocation to reproduction and/or growth early in life. Thus, one question arising from the present study is why *D. magna* do not evolve towards higher endogenous dopamine levels. One potential explanation for this may be that population growth rate estimates based on the timing and fecundity of the first two clutches is not always an appropriate fitness measure [68]. For example, this measure does not consider offspring survival and reproduction, which is an additional component of maternal fitness. Elevated dopamine levels caused reduced offspring size, and this may have negative fitness effects at low food abundance due to the relatively lower feeding efficiency of small individuals (see above). Alternatively, there may be ecological costs of expressing high dopamine levels and hence rapid growth, due to biotic interactions that were not quantified in this study. Rapid growth can increase predation costs through higher risk-taking behaviour from increased feeding in the presence of predators [69,70], as well as increased parasitism costs due to fewer resources being allocated to disease resistance [71,72]. Thus,

future studies should evaluate to what extent such selective factors can shape the evolution of the dopamine signalling system.

In summary, we found that the sign of the reaction norm in response to food abundance did not depend on the exposure treatment for most traits. Indeed, we showed an increase in adult size at the expense of growth and reproduction at restricted food ration for all treatments. Despite this general trend, the slopes of the reaction norms depended on the exposure treatment, as resource allocation to maturation and reproduction increased under dopamine and bupropion exposure when food rations were restricted, resulting in the advanced timing of reproduction at the expense of adult size and offspring size. Accelerated life cycles in the dopamine and bupropion treatments in turn resulted in higher population growth rates compared with the control, without any costs to longevity. This boost in fitness from dopamine upregulation contradicts our prediction that controls would have the highest fitness from having evolved adaptive reaction norms to food abundance. Further understanding of the evolution of the dopamine signalling system may require alternative measures of fitness that incorporate any effects on offspring survival and reproduction, as well as evaluating the potential for interactive effects between dopamine and ecological factors (predation, parasitism) on fitness. Nonetheless, our findings emphasize the role of the dopamine system as regulator of trait responses to food abundance and demonstrate that low but environmentally relevant concentrations of bupropion can alter the life history of *D. magna*, with possible consequences to individual fitness.

Ethics. All applicable international, national and/or institutional guidelines for the care and use of animals were followed. This study does not contain any studies with human participants performed by any of the authors.

Data accessibility. The data supporting this paper are available in the electronic supplementary material.

Authors' contributions. S.I. participated in the design of the study, carried out the laboratory work, carried out the statistical analysis and drafted the manuscript; S.E. conceived the study, participated in the design of the study, participated in data analysis and critically revised the manuscript; V.L.B.J. and T.M.C. participated in the design of the study and critically revised the manuscript; M.G. participated in data analysis and critically revised the manuscript; A.G.A. and K.V.-J. carried out the UPLC-MS/MS analysis and participated in the drafting of the manuscript. All authors gave final approval for publication and agree to be held accountable for the work performed therein.

Competing interests. The authors declare that they have no competing interests.

Funding. This work was supported by the Norwegian University of Science and Technology (NTNU) and the Research Council of Norway through its Centres of Excellence funding scheme, project no. 223257/F50.

Acknowledgements. We want to thank the staff at our lab facility for technical assistance and senior engineer Dr. Susana V. Gonzalez for the valuable help in supervising the UPLC-MS/MS analysis that was conducted at the Mass Spectrometry Lab at the Faculty of Natural Sciences (NV), NTNU.

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
