## [Reviewer comments · Proceedings of the Royal Society B: Biological Sciences]

Review History

RSPB-2019-2967.R0 (Original submission)

Review form: Reviewer 1

Recommendation

Accept with minor revision (please list in comments)

Scientific importance: Is the manuscript an original and important contribution to its field?

Excellent

General interest: Is the paper of sufficient general interest?

Good

Quality of the paper: Is the overall quality of the paper suitable?

Excellent

Is the length of the paper justified?

Yes

Should the paper be seen by a specialist statistical reviewer?

No

Do you have any concerns about statistical analyses in this paper? If so, please specify them explicitly in your report.

No

It is a condition of publication that authors make their supporting data, code and materials available - either as supplementary material or hosted in an external repository. Please rate, if applicable, the supporting data on the following criteria.

Is it accessible?

Yes

Is it clear?

Yes

Is it adequate?

Yes

Do you have any ethical concerns with this paper?

No

Comments to the Author

Reviewer's response

In the manuscript „Dopamine mediates life history responses to food abundance in *Daphnia*“, Issa et al. investigate the effect of the neurotransmitter dopamine and the dopamine up-take inhibitor buprion on *D. magna* in two different food regimes. They found effects of all treatments on reaction norms of life history traits (especially in the low food regime). I really enjoyed reading this very well-written manuscript and would suggest acceptance after minor revisions. I am no statistician; therefore, I am not sure whether the authors used the appropriate statistics, although the description sound reliable and sound. Hopefully, one of the other reviewers is more experienced in this topic.

Minor issues:

Line 41: maintenance of what?

Line 74 to 75: “Buprion inhibits the neuronal reuptake of norepinephrine and dopamine, increasing their concentration in the synaptic cleft.” In which organism?

Line 100-101: How many individuals were cultivated in 2.5 l water? No more than 40-60 mothers should be kept in this volume. Otherwise crowding effects cannot be excluded.

Line 208: please exchange “plentiful” with “adequate” or “sufficient” or “excessive”

Line 210: sign: do you mean direction? If so, please exchange.

Results: Please indicate the significance of the effects by adding significance levels.

Figures: Please indicate whether the effects are significant by adding indicators to the figures (stars/letters)

Supplementary: I would not “hide” the analysis of dopamine/buprion in the supplementary but include it in material and methods, partly in results (only the concentrations that are now included in Tab. A2) and add a sentence in the discussion.

Review form: Reviewer 2

Recommendation

Reject – article is not of sufficient interest (we will consider a transfer to another journal)

Scientific importance: Is the manuscript an original and important contribution to its field?

Marginal

General interest: Is the paper of sufficient general interest?

Marginal

Quality of the paper: Is the overall quality of the paper suitable?

Acceptable

Is the length of the paper justified?

Yes

Should the paper be seen by a specialist statistical reviewer?

Yes

Do you have any concerns about statistical analyses in this paper? If so, please specify them explicitly in your report.

No

It is a condition of publication that authors make their supporting data, code and materials available - either as supplementary material or hosted in an external repository. Please rate, if applicable, the supporting data on the following criteria.

Is it accessible?

N/A

Is it clear?

N/A

Is it adequate?

N/A

Do you have any ethical concerns with this paper?

No

Comments to the Author

Issa et al. conducted a life history experiment with *Daphnia magna* exposed to two different food availabilities, factorially combined with the addition of either the neurotransmitter dopamine or the antidepressant bupropion. As bupropion is thought to elicit increases in dopamine levels, it was expected that both compounds have similar effects on life history traits of *D. magna*. The study appears to be conducted in a largely appropriate way under controlled laboratory conditions. I would like to mention that I am not an expert on the applied statistical data analyses and thus cannot evaluate this part of the paper.

In terms of results, the authors observe strong effects of food availability on a range of life history traits. This is neither new nor surprising in any way. Beyond these well-known effects of food restriction on life-history parameters of *D. magna*, I am not quite sure what the reader can learn from this study.

Overall, the study's implications remain rather unclear. The strongest differential effects of bupropion and dopamine appear in terms of longevity under dietary restriction, which is probably the least ecologically relevant of all determined life history parameters. In nature, virtually no *D. magna* individual will die of old age. Longevity thus is a life history parameter that does not underlie any selection in nature. I further cannot envision a naturally relevant scenario for high external concentrations of dissolved neurotransmitters in the natural environment of *D. magna*. An ecologically relevant scenario for this treatment is either poorly explained or non-existent. I also do not think that application of relatively high aqueous neurotransmitter concentrations will help to understand "the role of the dopamine system as regulator of trait responses" (line 336) in *D. magna*.

A technical side remark: The effective concentrations could not be determined, neither for dopamine nor bupropion due to extended storage (4 months, line 133) between collection and analysis. As stated by the authors in line 139, it is not clear whether the very low recovery rates were due to "degradation during storage" or were that low already in the assay itself.

Review form: Reviewer 3 (S Kruppert)

Recommendation

Accept with minor revision (please list in comments)

Scientific importance: Is the manuscript an original and important contribution to its field?

Excellent

General interest: Is the paper of sufficient general interest?

Excellent

Quality of the paper: Is the overall quality of the paper suitable?

Excellent

Is the length of the paper justified?

Yes

Should the paper be seen by a specialist statistical reviewer?

No

Do you have any concerns about statistical analyses in this paper? If so, please specify them explicitly in your report.

No

It is a condition of publication that authors make their supporting data, code and materials available - either as supplementary material or hosted in an external repository. Please rate, if applicable, the supporting data on the following criteria.

Is it accessible?

Yes

Is it clear?

Yes

Is it adequate?

No

Do you have any ethical concerns with this paper?

No

Comments to the Author

In this study the authors tested the effects of dopamine as well as bupropion on life-history responses of *D. magna* to food-availability. Bupropion is an antidepressant that works as dopamine (and norepinephrine) re-uptake inhibitor. Their findings suggest that dopamine is part of the neuronal pass-way that induces food-availability based phenotypic plastic responses in *Daphnia*. Furthermore, they can show that the release of antidepressants, like bupropion, disrupts *Daphnia*'s life- and reproduction cycle under low food conditions.

The authors present a well designed study with solid experiments and statistical analysis. The manuscript is well written and of appropriate length. In fact, I like the study a lot. However, I have a few minor concerns that I would like to see addressed as well as some suggestions to improve the manuscript language.

My biggest concern is with the presentation of the statistical results. Please include test results details in the results section of the manuscript in order to clarify statistically significant differences between your treatments. I neither doubt your methods of choice nor do I doubt your findings, but adequate presentation of test results is best practice for good reason. Somewhat related, I think it would improve your manuscript to include a short section in your discussion that specifically addresses the difference between food-availability caused phenotypic plasticity and general effects caused by shortage of food. Again, your data supports your hypothesis in this regard but I think it would increase comprehensibility by clarifying that the observed effects are not direct effects caused by food shortage or starvation.

Please check the data availability policy of this journal thoroughly, they may not accept the "available upon request" option. You can easily add a supplementary table with all your measurements.

My further suggestions:

Abstract:

l20 "...bupropion, an antidepressant released into aquatic environments." That sounds like the drug is directly released into water bodies. Please consider to rephrase.

l22 add 'the' between 'to' and 'effects'

l24-26 Consider to put the sentence that starts with "We discuss why..." at the end of the abstract.

keywords: if you can, add "*Daphnia*" and "dopamine" to your keywords

Introduction:

l34 substitute "propensity" with "ability"

l41 substitute "maintenance" with "self-preservation"

l42 you can use "self-preservation" again instead of "survival"

l46 "At the proximate level,..." I do not understand what you mean by that.

l57 delete "otherwise"

l60 "are" instead of "is"

l66 change "eliminated" to "released"

l72 "can" instead of "may"?

I suggest you include a citation of Weiss 2019 (10.3389/fnbeh.2018.00330) in your introduction since it is a good review for neuronal pass-ways involved in phenotypic plasticity, even though it focuses on inducible defenses.

Materials and Methods:

l165ff Start the section with your software of choice. This way readers are not confused about it

when you first mention the used packages.

Results:

1216 delete "both"

1226 either delete "age at" or change adjective from "earlier" to "lower"

1235 consider rephrasing to "In addition to the influence of DM at maturation,..."

Discussion:

1263-265 rephrase to "This trade-off between offspring size and number is due to the limitation of energy [53, 54]." or similar.

figure captions:

1505 and 510 rephrase to "... traits in *D. magna* in the dopamine, the bupropion and the control treatment." or similar.

Very nice study, congratulations.

Decision letter (RSPB-2019-2967.R0)

03-Apr-2020

Dear Miss Issa:

I am writing to inform you that your manuscript RSPB-2019-2967 entitled "Dopamine mediates life history responses to food abundance in *Daphnia*" has, in its current form, been rejected for publication in Proceedings B.

This action has been taken on the advice of referees, who have recommended that substantial revisions are necessary. With this in mind we would be happy to consider a resubmission, provided the comments of the referees are fully addressed. However please note that this is not a provisional acceptance.

Please note that this decision may (or may not) have taken into account confidential comments.

In your revision process, please take a second look at how open your science is; our policy is that *ALL* (maximally inclusive) data involved with the study should be made openly accessible, fully enabling re-use, replication and transparency-- see:

<https://royalsociety.org/journals/ethics-policies/data-sharing-mining/>

Insufficient sharing of data can delay or even cause rejection of a paper.

Sincerely,

Dr John Hutchinson, Editor

Associate Editor

Comments to Author:

Dear Authors,

Your manuscript has now been seen by three experts in your field. Two of the reviewers see much merit in both the topic and manuscript, but one reviewer remains wholly unconvinced of the manuscripts suitability and impact for publication in Proceedings. If you decide to revise your manuscript for Proceedings please make sure to answer all reviewer comments and especially address the issues of manuscript novelty and impact brought up by Reviewer number 2.

Reviewer(s)' Comments to Author:

Referee: 1

Comments to the Author(s)

Reviewer's response

In the manuscript „Dopamine mediates life history responses to food abundance in *Daphnia*”, Issa et al. investigate the effect of the neurotransmitter dopamine and the dopamine up-take inhibitor buprion on *D. magna* in two different food regimes. They found effects of all treatments on reaction norms of life history traits (especially in the low food regime). I really enjoyed reading this very well-written manuscript and would suggest acceptance after minor revisions. I am no statistician; therefore, I am not sure whether the authors used the appropriate statistics, although the description sound reliable and sound. Hopefully, one of the other reviewers is more experienced in this topic.

Minor issues:

Line 41: maintenance of what?

Line 74 to 75: “Buprion inhibits the neuronal reuptake of norepinephrine and dopamine, increasing their concentration in the synaptic cleft.” In which organism?

Line 100-101: How many individuals were cultivated in 2.5 l water? No more than 40-60 mothers should be kept in this volume. Otherwise crowding effects cannot be excluded.

Line 208: please exchange “plentiful” with “adequate” or “sufficient” or “excessive”

Line 210: sign: do you mean direction? If so, please exchange.

Results: Please indicate the significance of the effects by adding significance levels.

Figures: Please indicate whether the effects are significant by adding indicators to the figures (stars/letters)

Supplementary: I would not “hide” the analysis of dopamine/bupropion in the supplementary but include it in material and methods, partly in results (only the concentrations that are now included in Tab. A2) and add a sentence in the discussion.

Referee: 2

Comments to the Author(s)

Issa et al. conducted a life history experiment with *Daphnia magna* exposed to two different food availabilities, factorially combined with the addition of either the neurotransmitter dopamine or the antidepressant bupropion. As bupropion is thought to elicit increases in dopamine levels, it was expected that both compounds have similar effects on life history traits of *D. magna*. The study appears to be conducted in a largely appropriate way under controlled laboratory conditions. I would like to mention that I am not an expert on the applied statistical data analyses and thus cannot evaluate this part of the paper.

In terms of results, the authors observe strong effects of food availability on a range of life history traits. This is neither new nor surprising in any way. Beyond these well-known effects of food restriction on life-history parameters of *D. magna*, I am not quite sure what the reader can learn from this study.

Overall, the study's implications remain rather unclear. The strongest differential effects of bupropion and dopamine appear in terms of longevity under dietary restriction, which is probably the least ecologically relevant of all determined life history parameters. In nature, virtually no *D. magna* individual will die of old age. Longevity thus is a life history parameter that does not underlie any selection in nature. I further cannot envision a naturally relevant scenario for high external concentrations of dissolved neurotransmitters in the natural environment of *D. magna*. An ecologically relevant scenario for this treatment is either poorly explained or non-existent. I also do not think that application of relatively high aqueous neurotransmitter concentrations will help to understand “the role of the dopamine system as regulator of trait responses” (line 336) in *D. magna*.

A technical side remark: The effective concentrations could not be determined, neither for dopamine nor bupropion due to extended storage (4 months, line 133) between collection and analysis. As stated by the authors in line 139, it is not clear whether the very low recovery rates were due to “degradation during storage” or were that low already in the assay itself.

Referee: 3

Comments to the Author(s)

In this study the authors tested the effects of dopamine as well as bupropion on life-history responses of *D. magna* to food-availability. Bupropion is an antidepressant that works as dopamine (and norepinephrine) re-uptake inhibitor. Their findings suggest that dopamine is part of the neuronal pass-way that induces food-availability based phenotypic plastic responses in *Daphnia*. Furthermore, they can show that the release of antidepressants, like bupropion, disrupts *Daphnia*'s life- and reproduction cycle under low food conditions.

The authors present a well designed study with solid experiments and statistical analysis. The manuscript is well written and of appropriate length. In fact, I like the study a lot. However, I have a few minor concerns that I would like to see addressed as well as some suggestions to improve the manuscript language.

My biggest concern is with the presentation of the statistical results. Please include test results

details in the results section of the manuscript in order to clarify statistically significant differences between your treatments. I neither doubt your methods of choice nor do I doubt your findings, but adequate presentation of test results is best practice for good reason. Somewhat related, I think it would improve your manuscript to include a short section in your discussion that specifically addresses the difference between food-availability caused phenotypic plasticity and general effects caused by shortage of food. Again, your data supports your hypothesis in this regard but I think it would increase comprehensibility by clarifying that the observed effects are not direct effects caused by food shortage or starvation.

Please check the data availability policy of this journal thoroughly, they may not accept the "available upon request" option. You can easily add a supplementary table with all your measurements.

My further suggestions:

Abstract:

l20 "...burpotion, an antidepressant released into aquatic environments." That sounds like the drug is directly released into water bodies. Please consider to rephrase.

l22 add 'the' between 'to' and 'effects'

l24-26 Consider to put the sentence that starts with "We discuss why..." at the end of the abstract.

keywords: if you can, add "Daphnia" and "dopamine" to your keywords

Introduction:

l34 substitute "propensity" with "ability"

l41 substitute "maintenance" with "self-preservation"

l42 you can use "self-preservation" again instead of "survival"

l46 "At the proximate level,..." I do not understand what you mean by that.

l57 delete "otherwise"

l60 "are" instead of "is"

l66 change "eliminated" to "released"

l72 "can" instead of "may"?

I suggest you include a citation of Weiss 2019 (10.3389/fnbeh.2018.00330) in your introduction since it is a good review for neuronal pass-ways involved in phenotypic plasticity, even though it focuses on inducible defenses.

Materials and Methods:

l165ff Start the section with your software of choice. This way readers are not confused about it when you first mention the used packages.

Results:

l216 delete "both"

l226 either delete "age at" or change adjective from "earlier" to "lower"

l235 consider rephrasing to "In addition to the influence of DM at maturation,..."

Discussion:

l263-265 rephrase to "This trade-off between offspring size and number is due to the limitation of energy [53, 54]." or similar.

figure captions:

l505 and 510 rephrase to "... traits in *D. magna* in the dopamine, the bupropion and the control treatment." or similar.

Very nice study, congratulations.

Author's Response to Decision Letter for (RSPB-2019-2967.R0)

See Appendix A.

RSPB-2020-1069.R0

Review form: Reviewer 1

Recommendation

Accept as is

Scientific importance: Is the manuscript an original and important contribution to its field?

Good

General interest: Is the paper of sufficient general interest?

Excellent

Quality of the paper: Is the overall quality of the paper suitable?

Excellent

Is the length of the paper justified?

Yes

Should the paper be seen by a specialist statistical reviewer?

No

Do you have any concerns about statistical analyses in this paper? If so, please specify them explicitly in your report.

No

It is a condition of publication that authors make their supporting data, code and materials available - either as supplementary material or hosted in an external repository. Please rate, if applicable, the supporting data on the following criteria.

Is it accessible?

Yes

Is it clear?

Yes

Is it adequate?

Yes

Do you have any ethical concerns with this paper?

No

Comments to the Author

I am fine with all corrections the authors made. It is now a scientifically sound manuscript.

Review form: Reviewer 3

Recommendation

Accept as is

Scientific importance: Is the manuscript an original and important contribution to its field?

Excellent

General interest: Is the paper of sufficient general interest?

Excellent

Quality of the paper: Is the overall quality of the paper suitable?

Excellent

Is the length of the paper justified?

Yes

Should the paper be seen by a specialist statistical reviewer?

No

Do you have any concerns about statistical analyses in this paper? If so, please specify them explicitly in your report.

No

It is a condition of publication that authors make their supporting data, code and materials available - either as supplementary material or hosted in an external repository. Please rate, if applicable, the supporting data on the following criteria.

Is it accessible?

Yes

Is it clear?

Yes

Is it adequate?

Yes

Do you have any ethical concerns with this paper?

No

Comments to the Author

I am pleased to see that you considered my comments valuable and included most of them in your manuscript. I especially like your supporting information section now since it is of remarkable extent and detail. Also, I am more than satisfied with your responses regarding the comments of mine that you chose not to include in your final draft. There is however one thing that I would love to address again without making it a required revision: I suggested a short discussion about the "difference between food-availability caused phenotypic plasticity and general effects caused by shortage of food." Thank you for reminding me in the response letter that phenotypic plasticity is not necessarily adaptive and therefore, per definition, include phenotypic consequences of food shortage. I actually did not take that into consideration in my first review. However, your discussion is based on the assumption that the observed plasticity is adaptive (1 273). I agree, since this observed pattern changes with the addition of bupropion and dopamine. If the observed differences between high and low food availability were an effect solely based on the energy available to the organism and not influenced by e.g. behavior or physiology, there would be no difference between the control and the experimental treatments,

since they all have the same amount of energy available. I hope that I am correct if I conclude that the observations thus display an 'adaption' to low food concentrations rather than a mere 'energy equation' (which both would qualify for phenotypic plasticity, which you pointed out correctly). My original comment was the suggestion to clarify that the observed differences between low and high food conditions are an adaptive response and not a direct effect solely due to differences in energy level. I hope I was able to explain the idea of my original comment and that the two different theoretical phenomena that I discuss are distinguishable. Please contact me if not or if you want to discuss this further since I think it is an interesting point. But to finally answer your question in the response letter: No, I do not think that this is crucial. Your manuscript is sound without it. This comment of mine was just a suggestion aiming for a slight increase of comprehensiveness for audience from other fields.

Decision letter (RSPB-2020-1069.R0)

04-Jun-2020

Dear Miss Issa

I am pleased to inform you that your manuscript RSPB-2020-1069 entitled "Dopamine mediates life history responses to food abundance in *Daphnia*" has been accepted for publication in Proceedings B. Congratulations!!

The referee(s) have recommended publication, but also suggest some minor revisions to your manuscript. Therefore, I invite you to respond to the referee(s)' comments and revise your manuscript. Because the schedule for publication is very tight, it is a condition of publication that you submit the revised version of your manuscript within 7 days. If you do not think you will be able to meet this date please let us know.

One reviewer has a final request but it is not deemed mandatory. However, we wish you to take it very seriously and if at all possible include it in the revised MS. It does seem that it would only help the paper be better.

- 1) A text file of the manuscript (doc, txt, rtf or tex), including the references, tables (including captions) and figure captions. Please remove any tracked changes from the text before submission. PDF files are not an accepted format for the "Main Document".

2) A separate electronic file of each figure (tiff, EPS or print-quality PDF preferred). The format should be produced directly from original creation package, or original software format. PowerPoint files are not accepted.

3) Electronic supplementary material: this should be contained in a separate file and where possible, all ESM should be combined into a single file. All supplementary materials accompanying an accepted article will be treated as in their final form. They will be published alongside the paper on the journal website and posted on the online figshare repository. Files on figshare will be made available approximately one week before the accompanying article so that the supplementary material can be attributed a unique DOI.

Sincerely,

Dr John Hutchinson, Editor

Associate Editor
Comments to Author:
Dear Authors,

Your paper has now been reassessed by two past reviewers (and myself). We all find the manuscript much improved and commend the authors on a fine job of answering the reviewer queries and comments. We thank the authors for submitting their paper here and wish them the best with final revisions.

Reviewer(s)' Comments to Author:

Referee: 3

Comments to the Author(s).

I am pleased to see that you considered my comments valuable and included most of them in your manuscript. I especially like your supporting information section now since it is of remarkable extent and detail. Also, I am more than satisfied with your responses regarding the comments of mine that you chose not to include in your final draft. There is however one thing that I would love to address again without making it a required revision: I suggested a short discussion about the "difference between food-availability caused phenotypic plasticity and general effects caused by shortage of food." Thank you for reminding me in the response letter that phenotypic plasticity is not necessarily adaptive and therefore, per definition, include phenotypic consequences of food shortage. I actually did not take that into consideration in my first review. However, your discussion is based on the assumption that the observed plasticity is adaptive (l 273). I agree, since this observed pattern changes with the addition of bupropion and dopamine. If the observed differences between high and low food availability were an effect solely based on the energy available to the organism and not influenced by e.g. behavior or physiology, there would be no difference between the control and the experimental treatments, since they all have the same amount of energy available. I hope that I am correct if I conclude that the observations thus display an 'adaption' to low food concentrations rather than a mere 'energy equation' (which both would qualify for phenotypic plasticity, which you pointed out correctly). My original comment was the suggestion to clarify that the observed differences between low and high food conditions are an adaptive response and not a direct effect solely due to differences in energy level. I hope I was able to explain the idea of my original comment and that the two different theoretical phenomena that I discuss are distinguishable. Please contact me if not or if you want to discuss this further since I think it is an interesting point. But to finally answer your question in the response letter: No, I do not think that this is crucial. Your manuscript is sound without it. This comment of mine was just a suggestion aiming for a slight increase of comprehensiveness for audience from other fields.

Referee: 1

Comments to the Author(s).

I am fine with all corrections the authors made. It is now a scientifically sound manuscript.

Author's Response to Decision Letter for (RSPB-2020-1069.R0)

See Appendix B.

Decision letter (RSPB-2020-1069.R1)

10-Jun-2020

Dear Miss Issa

I am pleased to inform you that your manuscript entitled "Dopamine mediates life history responses to food abundance in *Daphnia*" has been accepted for publication in Proceedings B.

Open Access

Paper charges

Sincerely,

Proceedings B

Appendix A

Title: Dopamine mediates life history responses to food abundance in *Daphnia*

Semona Issa^{1*}, Marlène Gamelon¹, Tomasz Maciej Ciesielski², Kristine Vike-Jonas³,
Alexandros G. Asimakopoulos³, Veerle L. B. Jaspers², Sigurd Einum¹

Manuscript revision

Authors: We thank the reviewers and editor for their constructive comments on our manuscript. We have addressed all the comments as indicated in blue and we have changed the manuscript accordingly as indicated by tracked changes in the revised manuscript (line numbers given below). We believe that our revisions have significantly improved the quality of our manuscript and hope that the revised version will meet your approval.

Referee: 1

Comments to the Author(s)

In the manuscript „Dopamine mediates life history responses to food abundance in *Daphnia*”, Issa et al. investigate the effect of the neurotransmitter dopamine and the dopamine up-take inhibitor buprion on *D. magna* in two different food regimes. They found effects of all treatments on reaction norms of life history traits (especially in the low food regime). I really enjoyed reading this very well-written manuscript and would suggest acceptance after minor revisions. I am no statistician; therefore, I am not sure whether the authors used the appropriate statistics, although the description sound reliable and sound. Hopefully, one of the other reviewers is more experienced in this topic.

A: We thank the reviewer for their positive evaluation of our manuscript.

Minor issues:

- Line 41: maintenance of what?

A: Maintenance refers to somatic maintenance. This information is now included in the manuscript (line 42).

- Line 74 to 75: “Buprion inhibits the neuronal reuptake of norepinephrine and dopamine, increasing their concentration in the synaptic cleft.” In which organism?
A: Bupropion inhibits the neuronal reuptake of norepinephrine and dopamine in any organism whose nervous system contains the neurotransmitters dopamine and norepinephrine. Most animal species synthesize dopamine (we have added this information on line 51), so the mode of action of bupropion on dopamine is widely observed across animal taxa.
- Line 100-101: How many individuals were cultivated in 2.5 l water? No more than 40-60
A: A maximum of 30 individuals were kept in this volume. We have added this information on line 102.
- Line 208: please exchange “plentiful” with “adequate” or “sufficient” or “excessive”
A: The change has been made (lines 215-216).
- Line 210: sign: do you mean direction? If so, please exchange.
A: We prefer to keep the term “sign” as it is the correct mathematical term to distinguish between a positive and a negative slope of a reaction norm. For the purpose of clarity, we have added the terms “(i.e. positive vs. negative slope)” and “steepness” on lines 218-219.
- Results: Please indicate the significance of the effects by adding significance levels.
Figures: Please indicate whether the effects are significant by adding indicators to the figures (stars/letters)
A: We have indicated the significance of the effects by adding i) p-values in the text throughout the results section (please see the revised manuscript with tracked changes); ii) indicators in the form of letters to figures 1 and 2.
- Supplementary: I would not “hide” the analysis of dopamine/buprion in the supplementary but include it in material and methods, partly in results (only the concentrations that are now included in Tab. A2) and add a sentence in the discussion.

A: We thank the reviewer for their observation. We have now briefly mentioned the sample preparation methods used and the subsequent analytical determination by UPLC–MS/MS (lines 136-140). The detailed description of the chemical analysis of dopamine and bupropion is quite long and likely not of particular interest for most readers of this journal (biologists). We therefore choose to let this section remain in the supplementary material for the purpose of brevity, while still being easily accessible for those interested. The results from this analysis and their discussion (lines 142-145) were placed in the materials and methods section in order to focus on treatment effects in the results and discussion sections.

Referee: 2

Comments to the Author(s)

Issa et al. conducted a life history experiment with *Daphnia magna* exposed to two different food availabilities, factorially combined with the addition of either the neurotransmitter dopamine or the antidepressant bupropion. As bupropion is thought to elicit increases in dopamine levels, it was expected that both compounds have similar effects on life history traits of *D. magna*. The study appears to be conducted in a largely appropriate way under controlled laboratory conditions. I would like to mention that I am not an expert on the applied statistical data analyses and thus cannot evaluate this part of the paper.

In terms of results, the authors observe strong effects of food availability on a range of life history traits. This is neither new nor surprising in any way. Beyond these well-known effects of food restriction on life-history parameters of *D. magna*, I am not quite sure what the reader can learn from this study.

A: We thank the reviewer for their assessment and observations. While we agree that effects of food restriction on life-history parameters of *D. magna* (and other organisms) are well known, this is not the focus of this study. Rather, this study shows how the strength of these responses depends on concentrations of substances that influence the strength of neural signalling. In other words, we assess the underlying molecular mechanisms shaping phenotypic plasticity in response to food availability, with focus on the dopamine signalling

pathway. We believe that this is quite strongly emphasised in all parts of the manuscript, including title, abstract, introduction, results and discussion.

Overall, the study's implications remain rather unclear. The strongest differential effects of bupropion and dopamine appear in terms of longevity under dietary restriction, which is probably the least ecologically relevant of all determined life history parameters. In nature, virtually no *D. magna* individual will die of old age. Longevity thus is a life history parameter that does not underlie any selection in nature.

A: The reviewer focuses on the differential effects of bupropion and dopamine on longevity here. It may (or may not, probably very dependent on the ecological setting of the population in question) be true that few *D. magna* die of old age. However, the general pattern observed in this study is that bupropion and dopamine have parallel effects on reaction norms in a range of other traits, as predicted based on prior knowledge of their chemical actions on the dopamine signalling system. We also respectfully disagree with the reviewer's argument concerning longevity being the least ecologically relevant of all determined life history parameters. Longevity is a life history trait commonly used as a measure of investment in long-term somatic maintenance. According to the principle of allocation (Cody 1966) and disposable soma theories (Kirkwood 1977), reduced longevity is expected as a consequence of a greater allocation to reproduction and/or growth early in life. Therefore, longevity that provides a measure of performance in terms of survival (maintenance) appears to be a key late-life trait in such context. We made this point clear in the revised manuscript (lines 324-326).

I further cannot envision a naturally relevant scenario for high external concentrations of dissolved neurotransmitters in the natural environment of *D. magna*. An ecologically relevant scenario for this treatment is either poorly explained or non-existent. I also do not think that application of relatively high aqueous neurotransmitter concentrations will help to understand "the role of the dopamine system as regulator of trait responses" (line 336) in *D. magna*.

A: The assessment of the role of the dopamine system as regular of trait responses was done using two complementary approaches:

- i) By manipulating dopamine levels in the growth medium. We agree with the reviewer that there is no naturally relevant scenario for high external concentrations of dissolved neurotransmitters in the natural environment of *D. magna*. However, aqueous exposure to dopamine allows us to directly manipulate this compound in the organisms (we have clarified this on lines 110-111), and thus provides a powerful approach to study its role in shaping reaction norms (see Weiss et al. 2015, also mentioned on line 113).
- ii) By manipulating the concentration of bupropion, the dopamine reuptake inhibitor, well known to increase internal dopamine levels. Bupropion is an antidepressant that enters natural aquatic environments and is thus highly relevant, from an ecotoxicological perspective.

A technical side remark: The effective concentrations could not be determined, neither for dopamine nor bupropion due to extended storage (4 months, line 133) between collection and analysis. As stated by the authors in line 139, it is not clear whether the very low recovery rates were due to “degradation during storage” or were that low already in the assay itself.

A: Yes, as already explained in the manuscript we are aware of this. Irrespective of this, the exposure concentrations were still high enough for treatments to induce significant biological effects in terms of observed changes in responses to food availability. By showing that manipulation of dopamine levels changes trait responses to food availability, we show that dopamine is a regulator of these responses.

Referee: 3

Comments to the Author(s)

In this study the authors tested the effects of dopamine as well as bupropion on life-history responses of *D. magna* to food-availability. Bupropion is an antidepressant that works as dopamine (and norepinephrine) re-uptake inhibitor. Their findings suggest that dopamine is part of the neuronal pass-way that induces food-availability based phenotypic plastic

responses in *Daphnia*. Furthermore, they can show that the release of antidepressants, like bupropion, disrupts *Daphnia*'s life- and reproduction cycle under low food conditions. The authors present a well designed study with solid experiments and statistical analysis. The manuscript is well written and of appropriate length. In fact, I like the study a lot. However, I have a few minor concerns that I would like to see addressed as well as some suggestions to improve the manuscript language.

A: We warmly thank the reviewer for their positive evaluation of our paper.

- My biggest concern is with the presentation of the statistical results. Please include test results details in the results section of the manuscript in order to clarify statistically significant differences between your treatments. I neither doubt your methods of choice nor do I doubt your findings, but adequate presentation of test results is best practice for good reason.

A: Following the reviewer's suggestion, we have indicated the significance of the effects by adding indicators in the form of letters to figures 1 and 2. The significance of the effects were also included in the results section by adding p-values.

- Somewhat related, I think it would improve your manuscript to include a short section in your discussion that specifically addresses the difference between food-availability caused phenotypic plasticity and general effects caused by shortage of food. Again, your data supports your hypothesis in this regard but I think it would increase comprehensibility by clarifying that the observed effects are not direct effects caused by food shortage or starvation.

A: We are sorry but not sure if we understand this comment. Effects of shortage of food on phenotype (compared to when food is in excess) is per definition phenotypic plasticity. If this is an important point, we hope the reviewer could explain more in detail.

- Please check the data availability policy of this journal thoroughly, they may not accept the "available upon request" option. You can easily add a supplementary table with all your measurements.

A: We thank the reviewer for their observation. All our measurements have been included in supporting information as Tables A8, A9 and A10.

My further suggestions:

Abstract:

- l20 "...burpopion, an antidepressant released into aquatic environments." That sounds like the drug is directly released into water bodies. Please consider to rephrase.

A: We have rephrased on lines 20-21.

- l22 add 'the' between 'to' and 'effects'

A: The change has been made (line 22).

- l24-26 Consider to put the sentence that starts with "We discuss why..." at the end of the abstract.

A: The change has been made (lines 28-29).

- keywords: if you can, add "Daphnia" and "dopamine" to your keywords

A: We prefer not to add "*Daphnia*" and "dopamine" as keywords as they are already in the manuscript title. Of course, we are willing to do it if the editor judges it necessary.

Introduction:

- l34 substitute "propensity" with "ability"

A: We prefer to keep the term "propensity" instead of "ability", as the latter makes phenotypic plasticity sound like it is always a wanted response, i.e. that it is adaptive in terms of increasing fitness. However, phenotypic plasticity may also be non-adaptive.

Propensity is then a better and more inclusive term to use.

- l41 substitute "maintenance" with "self-preservation"

A: We have substituted "maintenance" with "somatic maintenance" (line 42) as it is a well-established term used in the ecological/physiological literature.

- l42 you can use "self-preservation" again instead of "survival"

A: We have substituted "survival" with "somatic maintenance (survival)" (line 43).

- l46 "At the proximate level,..." I do not understand what you mean by that.

A: We have now clarified by substituting "At the proximate level" with "At the molecular level" (line 47).

- l57 delete "otherwise"

A: The change has been made (line 60).

- l60 "are" instead of "is"

A: The change has been made (line 62).

- l66 change "eliminated" to "released"

A: We prefer to keep the term "eliminated" as it is the more correct term here since it refers to humans and drugs are rather eliminated by excretion, metabolism, etc. than released.

- l72 "can" instead of "may"?

A: The change has been made (line 74).

- I suggest you include a citation of Weiss 2019 (10.3389/fnbeh.2018.00330) in your introduction since it is a good review for neuronal pass-ways involved in phenotypic plasticity, even though it focuses on inducible defenses.

A: We have added a reference of Weiss 2019 in the introduction section (lines 49-50).

Materials and Methods:

- l165ff Start the section with your software of choice. This way readers are not confused about it when you first mention the used packages.

A: We have deleted "in R v. 3.5.2" on line 193 and have added the following sentence "All statistical analyses and graphic illustrations were performed in R v. 3.5.2 [39]." on line 172.

Results:

- l216 delete "both"

A: The change has been made (line 225).

- I226 either delete "age at" or change adjective from "earlier" to "lower"

A: The change has been made (line 236).

- I235 consider rephrasing to "In addition to the influence of DM at maturation,..."

A: The change has been made (lines 246-248).

Discussion:

- I263-265 rephrase to "This trade-off between offspring size and number is due to the limitation of energy [53, 54]." or similar.

A: The change has been made (lines 275-277).

figure captions:

- I505 and 510 rephrase to "... traits in *D. magna* in the dopamine, the bupropion and the control treatment." or similar.

A: The change has been made (lines 523-526 and 528-531).

Very nice study, congratulations.

A: We thank the reviewer for their positive evaluation of our manuscript.

Appendix B

Title: Dopamine mediates life history responses to food abundance in *Daphnia*

Semona Issa^{1*}, Marlène Gamelon¹, Tomasz Maciej Ciesielski², Kristine Vike-Jonas³,
Alexandros G. Asimakopoulos³, Veerle L. B. Jaspers², Sigurd Einum¹

Manuscript revision

Authors: We thank the reviewers, associate editor and editor for their constructive comments on our manuscript. We have addressed all the comments as indicated in blue and we have changed the manuscript accordingly as indicated by tracked changes in the revised manuscript (line numbers given below). We believe that our revisions have significantly improved the quality of our manuscript and hope that the revised version will meet your approval.

Associate Editor

Comments to Author:

Dear Authors,

Your paper has now been reassessed by two past reviewers (and myself). We all find the manuscript much improved and commend the authors on a fine job of answering the reviewer queries and comments. We thank the authors for submitting their paper here and wish them the best with final revisions.

A: We thank the associate editor for their positive evaluation of our efforts and manuscript.

Reviewer(s)' Comments to Author:

Referee: 3

Comments to the Author(s).

I am pleased to see that you considered my comments valuable and included most of them in your manuscript. I especially like your supporting information section now since it is of remarkable extent and detail. Also, I am more than satisfied with your responses

regarding the comments of mine that you chose not to include in your final draft. There is however one thing that I would love to address again without making it a required revision: I suggested a short discussion about the "difference between food-availability caused phenotypic plasticity and general effects caused by shortage of food." Thank you for reminding me in the response letter that phenotypic plasticity is not necessarily adaptive and therefore, per definition, include phenotypic consequences of food shortage. I actually did not take that into consideration in my first review. However, your discussion is based on the assumption that the observed plasticity is adaptive (l 273). I agree, since this observed pattern changes with the addition of bupropion and dopamine. If the observed differences between high and low food availability were an effect solely based on the energy available to the organism and not influenced by e.g. behavior or physiology, there would be no difference between the control and the experimental treatments, since they all have the same amount of energy available. I hope that I am correct if I conclude that the observations thus display an 'adaption' to low food concentrations rather than a mere 'energy equation' (which both would qualify for phenotypic plasticity, which you pointed out correctly). My original comment was the suggestion to clarify that the observed differences between low and high food conditions are an adaptive response and not a direct effect solely due to differences in energy level. I hope I was able to explain the idea of my original comment and that the two different theoretical phenomena that I discuss are distinguishable. Please contact me if not or if you want to discuss this further since I think it is an interesting point. But to finally answer your question in the response letter: No, I do not think that this is crucial. Your manuscript is sound without it. This comment of mine was just a suggestion aiming for a slight increase of comprehensiveness for audience from other fields.

A: We thank the reviewer for their positive evaluation of our efforts and manuscript. We have now added this information on lines 282-286.

Referee: 1

Comments to the Author(s).

I am fine with all corrections the authors made. It is now a scientifically sound manuscript.

A: We thank the reviewer for their positive evaluation of our efforts and manuscript.